# A Model Free Control Based on Machine Learning for Energy Converters in an Array

**Simon Thomas** [1,*]**, Marianna Giassi** [1] **, Mikael Eriksson** [1]**, Malin Göteman** [1]**, Jan Isberg** [1]**, Edward Ransley** [2]**, Martyn Hann** [2] **and Jens Engström** [1]

1   Ångströmlaboratoriet, Division of Electricity, Uppsala University, Lägerhyddsvägen 1,
    75237 Uppsala, Sweden; Marianna.Giassi@angstrom.uu.se (M.G.);
    Mikael.Eriksson@angstrom.uu.se (M.E.); Malin.Goteman@angstrom.uu.se (M.G.);
    Jan.Isberg@angstrom.uu.se (J.I.); Jens.Engstrom@angstrom.uu.se (J.E.)
2   School of Engineering, University of Plymouth, Drake Circuit, Plymouth PL4 8AA, UK;
    Edward.Ransley@plymouth.ac.uk (E.R.); Martyn.Hann@plymouth.ac.uk (M.H.)
*   Correspondence: simon.thomas@angstrom.uu.se

**Abstract:** This paper introduces a machine learning based control strategy for energy converter arrays designed to work under realistic conditions where the optimal control parameter can not be obtained analytically. The control strategy neither relies on a mathematical model, nor does it need a priori information about the energy medium. Therefore several identical energy converters are arranged so that they are affected simultaneously by the energy medium. Each device uses a different control strategy, of which at least one has to be the machine learning approach presented in this paper. During operation all energy converters record the absorbed power and control output; the machine learning device gets the data from the converter with the highest power absorption and so learns the best performing control strategy for each situation. Consequently, the overall network has a better overall performance than each individual strategy. This concept is evaluated for wave energy converters (WECs) with numerical simulations and experiments with physical scale models in a wave tank. In the first of two numerical simulations, the learnable WEC works in an array with four WECs applying a constant damping factor. In the second simulation, two learnable WECs were learning with each other. It showed that in the first test the WEC was able to absorb as much as the best constant damping WEC, while in the second run it could absorb even slightly more. During the physical model test, the ANN showed its ability to select the better of two possible damping coefficients based on real world input data.

**Keywords:** machine learning; wave energy; power take-off; artificial neural network; wave tank test; physical scale model; floating point absorber; damping; control; collaborative

## 1. Introduction

In order to make the power production more sustainable, a wide range of new power plant technologies are entering the energy sector. Compared to traditional energy carriers, sustainable resources often have a much lower power density. This can be tackled by designing smaller energy converters that are arranged in farms.

A major aspect for the success of a new electrical power technology are the costs, which should be close to or less than the costs of already available technologies. Until now, only a few sustainable energy sources have overcome this challenge, for example wind power. One way of lowering the energy costs is to increase the absorbed power using advanced control strategies. Control strategies like impedance matching [1] for Wave Energy Converters (WECs) can allow the converter to operate

at its theoretical maximum, but they are hard to implement in reality as they are not causal and it takes some effort to apply physical constraints to it. If working with constraints, Model-Predictive Control (MPC) is state of the art. Examples can be found in [2,3], but the quality of this strategy relies heavily on theoretical models and the ability to forecast. Especially when absorbing energy from fluids, the complex interactions make it very hard to develop reliable models. The authors own experiences in [4] using a physical WEC model showed that even small differences between physical device and theoretical calculations can lead to significant changes in power output and optimal operation. The robustness to model inaccuracies is an important characteristic of a control. New controls strategies, like the ones presented in [5] achieve results close to MPC, but are much more robust to model errors. This paper wants to go one step further, presenting a model-free control, that parametrizes itself during operation (online). Furthermore it should interact dynamically with other energy converters to improve the absorbed power making it a suitable control for farms with strong coupling between the converters.

In recent years machine learning (ML) has become popular in obtaining reasonable solutions for tasks which are otherwise hard to solve. It replaces analytical methods by sample-intensive optimization processes, it is therefore important that the costs of a single sample are low, making ML a good choice when huge data sets are available or can be obtained quickly. An example showing the strength of model-free ML can be found in [6], where the control of a fish-like robot was parametrized using an online genetic-algorithm on a model in a water tank.

Within the field of ML, artificial neural networks (ANN) have produced significant progress in recent years. They achieved remarkable results in different areas like speech recognition [7], picture analysis [8], Artificial Intelligence [9] and non-linear control [10]. The most important basic learning concepts are supervised and reinforcement learning, but none of them fulfill the following needs:

- For reinforcement learning a cost function is missing that describes which energy output has to be seen as positive for a specific state.
- For traditional Supervised learning the training data is missing as the control strategy should learn "on the fly".

Artificial neural networks are relatively rarely used for the control of energy converters. In the following a few examples from the area of wave energy converters are given: In [11] a genetic algorithm was used offline to parametrize an artificial neural oscillator to find the optimal latching times for a wave energy absorber. The problem of getting an accurate model is tackled in [12] with the help of ANN: The network is trained to mimic the behaviour of a WEC, the so obtained model is then used by a conventional control algorithm to predict the best working state for the real buoy. In [13] an approach is presented that learns "on-the-fly" using a two step approach: An initial network categorizes the sea states and a second evaluates the optimal parameters for each state.

In this paper the on-the-fly (or online) learning strategy is adapted to make use of the array configuration in which the energy converter will operate. We extend the supervised learning approach by generating the reference pattern during the learning phase. Therefore two or more energy absorbers, of which at least one has to be controlled by the artificial neural network presented in this paper, have to be placed so that they are effected by the energy medium at the same time. During operation all energy absorbers apply their control strategy and monitor their absorbed energy; the control strategy with the highest energy absorption is then used to train the network.

This approach differs from the similar Ensemble learning, as it does not try to find the optimal model for each state, but use it to learn a network. The aim of this approach is to get a single neural network handling all cases, instead of a set of models which are specialized in different fields; this gives a big degree of freedom, as the process is not divided into learning and operating. Also the approach is strictly speaking a multi-agent system, it should be classified as a parallel single-agent system, due to the neglected interactions between the devices.

As in this strategy several energy converters are helping each other (or at least they help all learnable algorithms in the group) to solve a common task; this paper will refer to this strategy as Collaborative Learning (CL), analogue to the teaching method in schools [14].

The CL strategy will be demonstrated on a WEC array (set up see Figure 1).

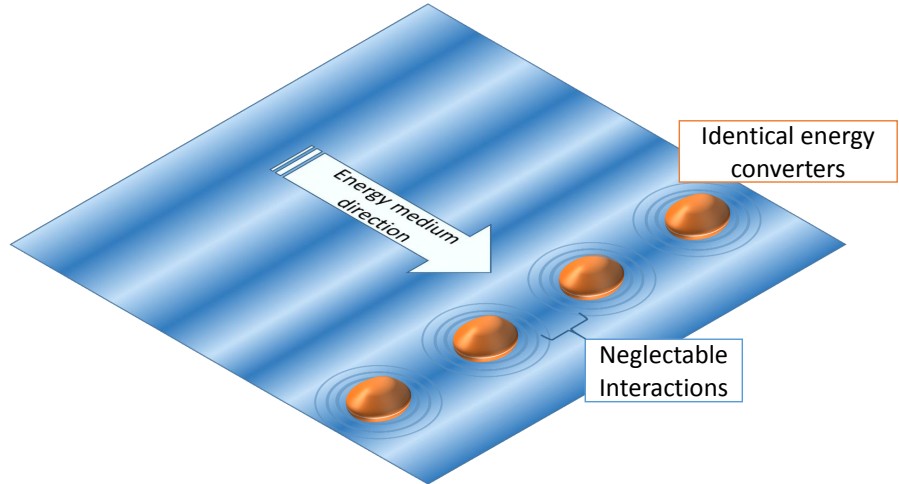

**Figure 1.** Sketch of an energy converter array that can be used for CL.

WECs are used to extract energy from ocean waves, and are therefore ment to play an important role in providing a sustainable energy source that can be more accurately forecasted than wind and solar energy [15]. From the Edinburgh Duck, one of the earliest and most well known modern WEC, a lot of different designs have been developed, but in general they can be categorized in three types: Attenuator, Point absorber and Terminators [16]. The CL strategy will work for any type as long as it is controllable. However this paper will concentrate on a direct driven point absorber with an electrical linear generator, similar to the devices which are built and operated by Uppsala University on the west coast of Sweden [17]. In contrast to these real existing devices, here it is assumed that the generator damping can be adjusted by choppering the coils [18]. This is a relatively cheap and effective way to control a WEC and may be implemented in existing devices without large modifications.

The next chapter starts with the detailed presentation of the collaborative control strategy. It then continues with the description of the physical scale model and the mathematical model behind the numerical simulation. Chapter 3 is about how the experiments were performed and shows the results, which will then be discussed in the fourth chapter. The last part is an outlook what might be done in the future.

## 2. Collaborative Learning

To focus on the CL strategy instead of the artificial neural network itself, only well established approaches for the network design are used. For systems with temporal dynamic behaviour two network concepts are common in the literature:

- Recurrent neural networks (RNN), where the connection between units form a directed cycle. A popular subset of RNNs are long-short-term memory networks. For RNN the computational costs are higher than for feed-forward-networks.
- In feed forward networks the connection between units can not form a cycle, the information passes only in one direction. To give them the ability to memorize, the input layer has to be extended with units getting a time-delayed input signal.

Due to the reduced computational complexity a feed-forward-network is used. The weights were updated using back propagation. For the input, at least one signal has to be related to the amount of

currently absorbable energy. The number of units in the input layer depends on the length of the series of previous input values. The output neuron(s) is/are the state(s) of the system that can be controlled.

### 2.1. Collaborative Learning

For training the network uses a special form of supervised learning. In the classic form of supervised learning a teacher knows the correct input and output data. The network changes its internal weights so that the error between network-output and reference data for a specific input pattern decreases. Therefore during the learning process the output error is back-propagated through the network to adjust the weights between the units in all layers.

In this case there is no reference data, and it can not even be judged if the output for a given input is beneficial. But there are several energy converters in a row, which are all affected at the same time. With these prerequisites, the approach is to apply different control strategies to each converter, of which at least one is the artificial neural CL network. The power output and the applied damping are logged for each converter. After a specified time frame $t_{set}$, which consists of several pairs of measured input and applied output data (called a sample), the average energy absorption is calculated. The samples of a converter recorded within this time frame are called a sample set. The sample set of the converter that absorbs the highest amount of power is used to train the neural network. In the following the complete procedure is described step by step:

### 2.2. The CL Process

1.  Initial condition: Two or more identical energy converters, all applying different control strategies, of which at least one is the CL network, are placed so that they will be affected by the energy resource at the same time.
2.  The main program sends the "Start" message to all converters.
3.  The converters write continuously the sample data (values of the input and output units) in a log file.
4.  After a time period $t_{set}$, the main control sends the "Stop" message to all converters; in return the main program receives all sample sets.
5.  The CL algorithm calculates the average absorbed power for each converter and chooses the sample set of the converter with the highest absorption.
6.  This sample set's input and output is used to train the CL network.
7.  After the training is finished, the procedure repeats from step 2.

The process is visualized in Figure 2.

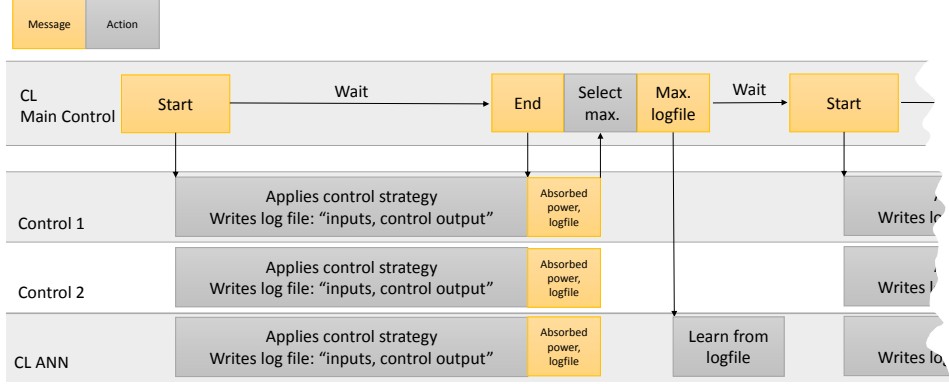

**Figure 2.** Block diagram of the CL control method.

The sampling frequency $f_f$ as well as the time interval $t_{set}$ have to be equal for all converters. If the controller focuses on the short or long term power absorption depends on the sample set duration:

A short duration will push the CL-network to maximize the current power absorption, without taking into account if the action that is taken now will also be beneficial for power absorption in the future. Under realistic conditions, the lower limit for $t_{set}$ is the time the controller needs to apply a specific parameter and get reliable measurements of how this effects the power absorption. Furthermore, when it can not be guaranteed that all converters are effected by the energy source at the same time, $t_{set}$ should be much higher than the time differences between the converters. $t_{set}$ should be chosen much smaller than a specific state of the energy resource (for example wind speed, sea state) typically lasts.

The weights were updated after each learning process, no batch learning was used. The learning rate depends on a manually set learning factor based on the current learning status and (with a smaller influence) a factor based on the absorbed power compared to the absorbed power of the neuronal network. Each sample set was learned directly after it was obtained, it was then stored in a pool of learning samples. Each time a new sample was created and used for training, several samples from the pool were trained additionally The algorithm ensured that each sample was 100 times used for training and that the samples from the pool were picked in random order to avoid overfitting. To fit in the existing WEC control framework for the wave tank model (see below) and ensure good performance a self-written ANN algorithm was used.

### 2.3. Evaluation

After the learning phase an ANN is evaluated with a second test data set. During the test session, the ANN calculates the output without information about the reference, i. e. without learning. The data is then compared with the reference. Training and test data should be slightly different to evaluate the network's ability to generalize.

### 3. Numerical and Experimental Set-Up

The type of point absorber used in this paper is inspired by the WECs developed by Uppsala University for the Lysekil project [17]. These wave converters consists of three parts: The float, here a cylindrical buoy, the Power-Take-Off (PTO), here an electrical linear generator with adjustable damping, and a line connecting the buoy and PTO, see also Figure 3.

The translator of the linear generator has a given stroke length $l_T$, so that the position of the translator $y$ has to be within the limits:

$$0 \leqq y \leqq l_T.$$

To protect the mechanical casing, springs with a length $l_c$ are mounted on the top and bottom to prevent the translator from hitting the end stops.

In contrast to the real existing generators, the converter used in this work is able to actively change the damping in short term, which is a simple way of controlling the WEC and optimize wave energy absorption for different sea states. For a real generator the damping $\gamma$ can only be adjusted within a range:

$$\gamma_{min} \leqq \gamma \leqq \gamma_{max}.$$

A load cell is placed between line and translator to measure the force in the line.

For collaborative learning two or more devices have to be in line parallel to the wave front. How the interaction between buoys influences the power absorption and optimal damping of WECs in wave farms is a topic of ongoing research, but most studies suggest that the influence is neglectable small if the WECs are in line to the wave front and have sufficient space in-between each other [19]. For now it is assume that the interaction between the WECs in our array will have no significant influence on the optimal damping and absorbed power.

### 3.1. Numerical Model

The simulation uses a two body model consisting of buoy, translator and a connecting line. As the control strategy itself does not care about the specific WEC the main goal was to achieve a fast

numerical model that captures the relevant behaviour qualitatively rather than a quantitative accurate simulation. Therefore the assumptions of linear potential theory is used, but in contrast to many other numerical models, this approach does not use boundary-integral equations methods (BIEMs) to calculate the hydrodynamic coefficients. Since point absorber have a small surface area, and linear potential theory assumes flat waves, the pressure over the surface is in this simulation approximated as being constant and the defraction of the waves is neglected. Furthermore the only motion of interest is in heave direction. These assumptions lead to a fast, but accurate model, which is not based on convolution terms.

As no numerical model is able to simulate all aspects of a physical test, the wave tank test is performed in addition to the numerical model focusing on how the algorithm handles measured data.

### 3.1.1. Heave Force

The force from the water surface in heave direction is, according to Bernoulli's principle, made of two forces, the potential force $F_{Bc}$ and the kinetic force $F_H$. Both depend on the surface elevation $h$ and the buoy's position $x$. For the heave force only the kinetic force is considered, the potential force is included in the buoyancy force. As long as the buoy's position is in the water and the surface level is rising upwards relative to the buoy's motion, the heave force is applied.

$$F_H = \begin{cases} A_B \rho (\dot{h} - \dot{x})^2 & , x < h \wedge \dot{h} - \dot{x} > 0 \\ 0 & , \text{else} \end{cases}, \tag{1}$$

where $\rho$ is the density of water and $A_B$ is the (wetted) area of the buoy parallel to the surface.

### 3.1.2. Buoy

The buoy is modelled as a linear mass-spring-damper system which is affected by the heave force $F_H$, the buoyancy force $F_{Bc} = \rho g A_B (h - x)$, the gravity force $F_{Bg} = m_B g$, with $g$ being the gravitational acceleration constant, the hydrodynamic damping force $F_{bd} = d_B \dot{x}$ and the line force $F_L$. The equation of motion for the buoy is

$$\ddot{x} = \frac{F_H + F_{Bc} + F_{Bg} + F_{bd}}{m_B + m_A},$$

with $m_B$ being the mass and $m_A$ being the added mass of the buoy, and $F_{bd}$ the damping of the buoy, motivated by the radiated wave of the buoy.

### 3.1.3. Line

The line, modelled as a spring-damper system, is only activated if the distance between buoy and translator is larger than the length of the line.

$$F_L = \begin{cases} c_R (x - y) + d_R (\dot{x} - \dot{y}) & , x > y \\ 0 & , \text{else} \end{cases}, \tag{2}$$

with $c_r$ being the stiffness and $d_R$ the damping of the line.

### 3.1.4. Translator

The translator is modelled as a linear mass-damper system, which is affected by the gravity force $F_{Tg}$ and the line force $F_L$. The damping force $F_{Td}$ is the product of the damping factor $\gamma$ and the translator's velocity; $\gamma$ is set by the control algorithm. So the force $F_T$ acting on the translator is:

$$F_T = F_{Tg} + F_{Tm} + F_{Td} - F_L, \tag{3}$$

with $F_{Tg} = m_T g$ and $F_{Tm} = m_T \ddot{y}$; $m_T$ is the mass of the translator. Furthermore an end stop force $F_{Stop}$ is introduced which consists of a spring and a position limit. The position limits are at the upper and lower end of the translator. The springs have a length of $l_s$ and are positioned at the end stops. They provide the spring force $F_{Ts}$.

$$F_{Stop} = \begin{cases} F_{Ts} & , l_s > y > 0 \vee l_T - l_s < y < l_T) \\ 0 & , \text{else} \end{cases}.$$

The equation of motion for the translator is:

$$\ddot{y} = \frac{F_{Tg} + F_{Td} - F_L + F_{Stop}}{m_T}. \tag{4}$$

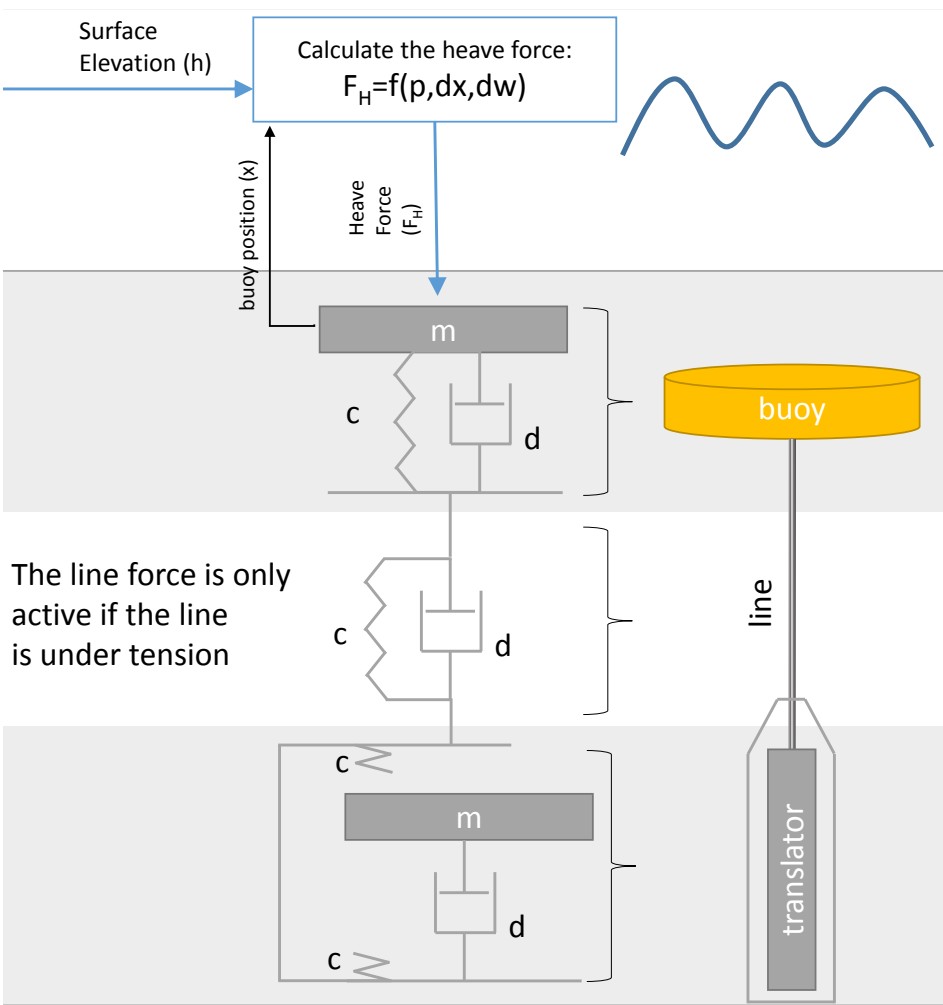

**Figure 3.** Block diagram of the model, c is the stiffness of the spring, d the damping coefficient and m the mass.

### 3.1.5. Parametrisation

The parameters of the simulations can be found in Table 1. They correspond to a 1:10 scale of the full scale WEC, which is similar to the device used in the wave tank test. The stiffness of the line was limited by the numerical stability of the simulation; the damping in the line was set so that there were no significant oscillations in the line. The added mass and damping of the buoy were chosen to match with data obtained from a decay test with the physical buoy model.

**Table 1.** Parameters.

| | | |
|---|---|---|
| translator | $m_T$ | 6 kg |
| | $l_T$ | 0.32 m |
| | $l_s$ | 0.08 m |
| buoy | $m_B$ | 5 kg |
| | $m_A$ | 0.3 kg |
| | $S_w$ | 0.2 m$^2$ |
| | $d_B$ | 50 Ns/m |
| line | $d_R$ | 84,000 $\frac{Ns}{m}$ |
| | $c_R$ | 40,000 $\frac{N}{m}$ |

### 3.2. Wave Tank Model

For the wave tank test an ellipsoidal buoy with a diameter of 0.5 m was used, which was connected with a line, guided by a pulley system, to the PTO. The PTO consisted of an electric linear motor that mimics a generator, which enables us to implement nearly all common control strategies. In this paper it is used to simulate a generator with constant damping. So the applied force is proportional to the measured speed of the translator:

$$F_{PTO} = \gamma \dot{y}.$$

A more detailed description of the set-up can be found in [4]. The force controller runs with a frequency of 100 Hz.

## 4. Experiments

The simulation and the wave tank test were designed to supplement each other. Both use similar 1:10 scale buoys and the same neural network. During the simulation the CL algorithm is training along with four constant damping controls.

In the tank test only two WECs can run at the same time, which leads to a slightly modified test: The sample sets of two constant damping controlled WECs are recorded and then the winning sampling set is fed to the neural network. This procedure is repeated for the training wave, but at this time, the network is not trained with the sample set but had to predict the winning damping.

### 4.1. Artificial Neural Network

The main objective with designing the artificial neural network was to achieve a fast learning process; especially for the experiments in a wave tank noticeable results should be seen in less than one hour. Complex deep networks that have achieved remarkable results in different areas, come with the cost of a long learning process which requires a big training data set. Finding the balance between pattern recognizing and fast learning progress, a network with eight hidden layers is chosen as the main network (CL-ANN1). Furthermore a second network (CL-ANN2) was applied in the simulation, using two hidden layers and a random generator, which added a random number to the output, giving it the ability to explore new damping coefficients outside its policy. A sketch of the set-up of both networks can be found in Figures 4 and 5. To speed up the learning process and get the samples sets randomized, each sample set is learned ten times in intervals of a few minutes.

The input has to be related to the current wave force acting on the WEC. This could be done by placing a wave measurement buoy in line with the WECs, but with an adequate accuracy the wave force can also be obtained by a load cell measuring the force in the line and subtracting the damping force of the generator. Inertia and wave interaction of the buoy still influence the rope force, but as they are similar for all WECs, it will not influence the learning process of the CL-network.

Real surface gravity waves used for wave power have frequencies lower than 0.5 Hz and according to the Nyquist-criteria the sampling frequency should then be higher than 1 Hz. We set it to $f_f = 1.5$ Hz (full scale). The number of input units was set to 5, as $\frac{1}{f_f}10 = 6.67\,s$ is a suitable window length. While the first input unit get the current force signal, the second get previous value (delayed by $\frac{1}{f_f}$) and so on.

The output of the network is the normalized damping $\gamma_{out}$, using the following formula to resize it to its actual value $\gamma$:

$$\gamma = (\gamma_{out}(\gamma_{max} - \gamma_{min})) + \gamma_{min}$$

A sample set duration was set to $t_{set} = 3\,s$ (full scale) in the simulation and to $t_{set} = 6\,s$ (full scale) in the wave tank test.

As activation function for the hidden layers the ramp (ReLU) is used, which is known to enable fast learning rate over layers [20], and a logistic function is applied for the output units. The bias and weights of all units are set randomly.

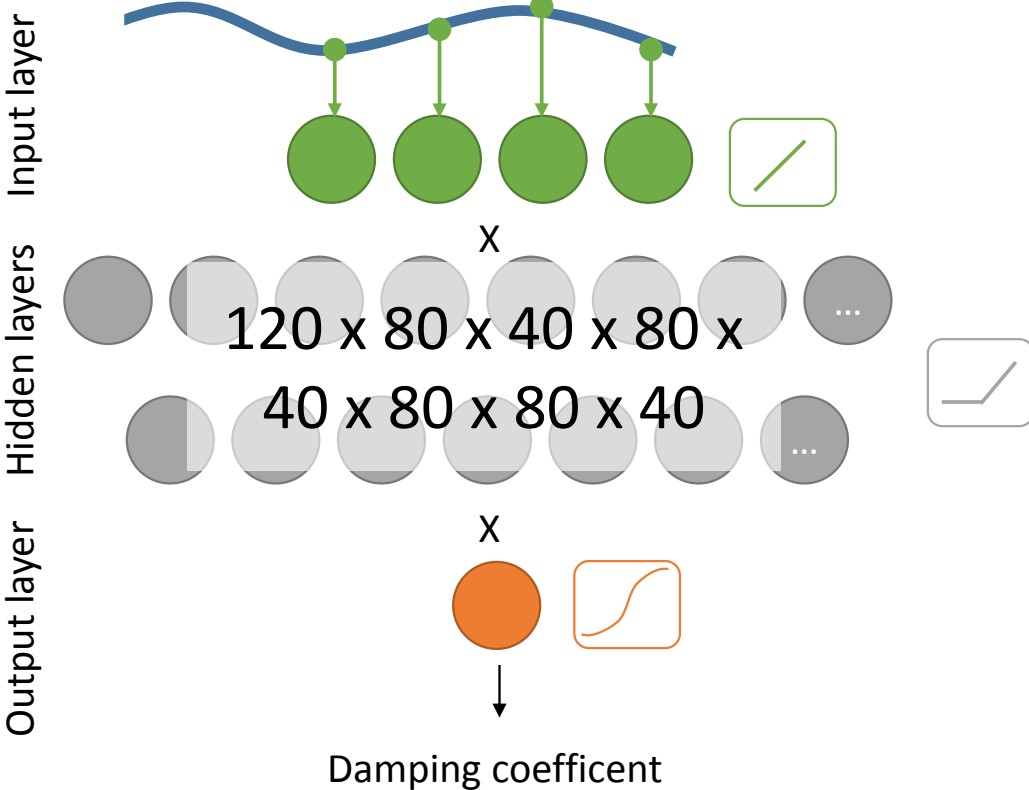

**Figure 4.** Number of layers and neurons for the collaborative learning network 1 (CL-ANN1).

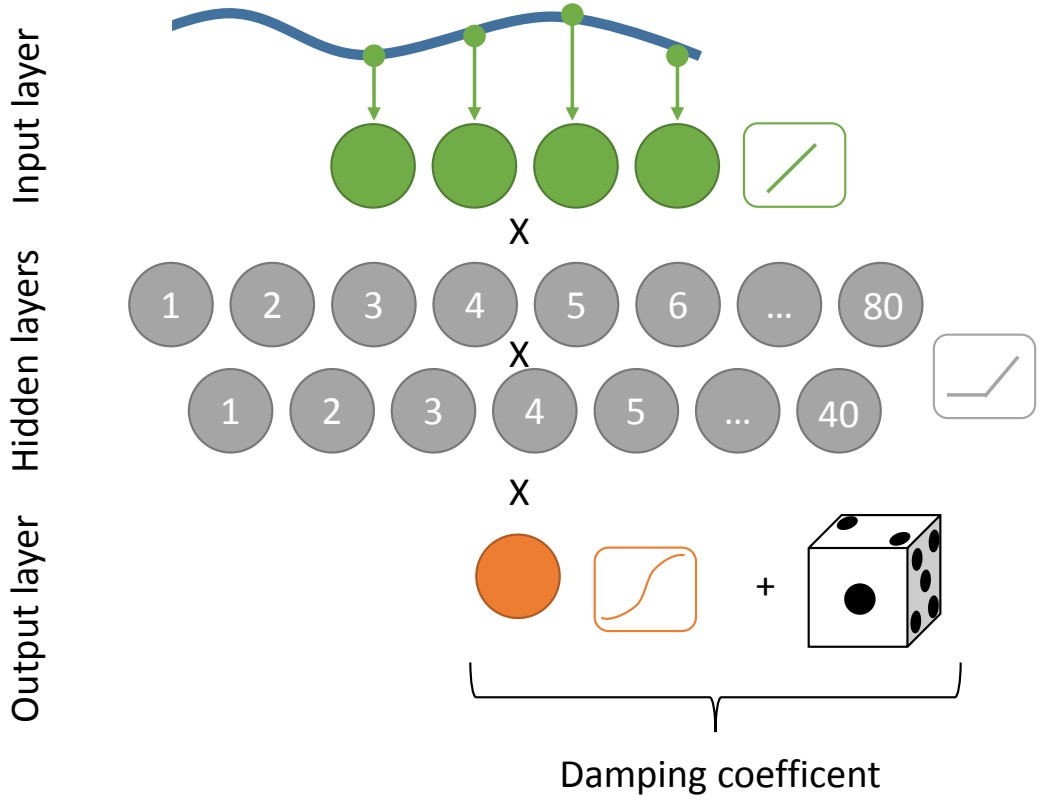

**Figure 5.** Number of layers and neurons for the collaborative learning network 2 (CL-ANN2).

### 4.2. Wave Data

The training wave data used for both - simulations and wave tank tests—is a 10 min (about half an hour in full scale) long medley consisting of 20 Bretschneider spectra sea states ranging from a full scale energy period of 3.5 s to 10.5 s and from a full scale significant wave height of 0.75 m to 3.25 m. The sequence was designed in such a way, that different sea states alternate, but at the same time it was ensured that the sea state will not change abruptly.

The test wave sequence is similar to the training data but shorter (3 min, 51 s in 1:10 scale) and consists of eight Bretschneider spectra which were not used during the training period, but are within the same range of energy period and wave height.

Both wave sequences can be seen in Figure 6.

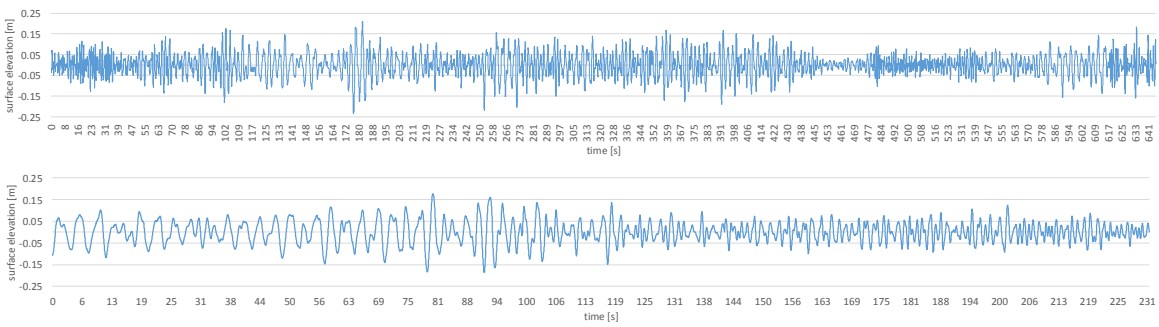

**Figure 6.** Training (**above**) and test (**below**) wave data in the applied 1:10 scale.

### 4.3. Simulation Test

Two simulations were done, in which the WECs were placed in a row perpendicular to the wave front and all hydrodynamic interactions between the buoys were neglected. The first simulation is called static, as among the five WECs which were simulated only one uses the CL-network; the four other WECs used a static constant damping with 200, 300, 400 and 500 Ns/m.

During the second simulation the focus was on the dynamic learning process between two CL-networks. To prevent both networks from doing the same, they were given different 'characters'. The first network (CL-ANN1) is slow learning and looks for the best control strategy in long term while the second network (CL-ANN2) has the task to explore new damping coefficients. All control algorithm are connected and supervised by the main program, that is starting and stopping the recording of the input and output data of each control.

Figure 7 shows the average absorbed power for each WEC during the test sequence.

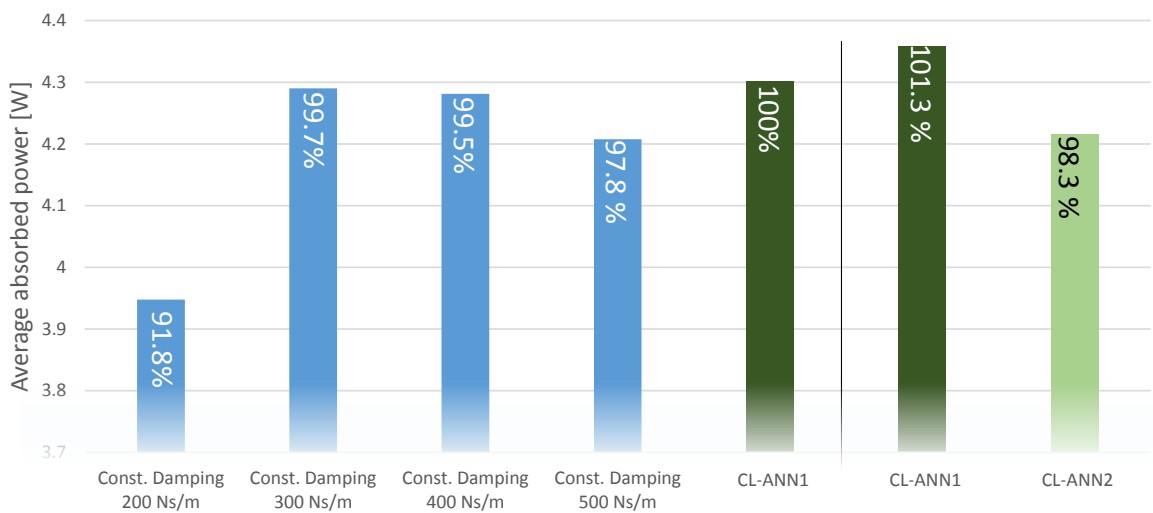

**Figure 7.** Average absorbed power for the test wave sequence in the simulations. CL-ANN1 indicates the 'deep' ANN, and CL-ANN2 the 'shallow' ANN.

### 4.4. Wave Tank Test

The physical test was performed in a 1:10 scale in the Ocean Basin of the COAST laboratory of Plymouth University. The test was limited to two running WECs at the same time. Both WECs used a constant damping control and were connected to the CL main program, which synchronized the recording of the data sets. Instead of learning the CL network online, a log file with the winning data set for each time period was written and used to train the CL network offline. Due to friction in the mechanical system the used dampings were much lower then in the simulation and lay on a very flat part of the logistics function, which was motivation to replace the logistic function for the output neurons of the CL network with the linear function.

The same procedure was implemented for the test wave sequence, but instead of learning the network, it was only fed with the input data and had to guess the corresponding damping factor.

This method had the disadvantage that the ANN was not active itself, but is a good measure of how well the network handles real world data: In Figure 8 the winning damping over the test data is plotted in comparison with the ANN output for the same data input.

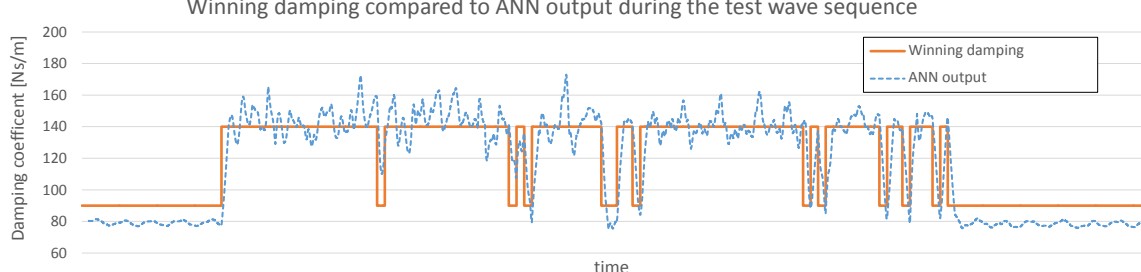

**Figure 8.** Winning damping of both constant damping controls (orange, solid line) compared to the output for the same input of the ANN (moving average, blue dotted line). To increase readability the ANN output is represented by a moving average of 100 values.

## 5. Discussion

### *5.1. Simulation*

The model is based on simplified hydrodynamic wave-body interactions. Therefore the magnitude of the in- and output values, the damping factors and the absorbed power, are briefly discussed.

The damping factors of the generator leading to the highest power absorptions in the numerical simulation (Figure 7) are very high; Previous work [4] suggested much lower optimal damping factors. This may be caused by some inaccurate hydrodynamic parameters (for example underestimation of hydrodynamical damping) due to the simplification made. However the power output is with about 4 W in the expected range [4], indicating that the hydrodynamic inaccuracies may have only small influence on the overall performance of the model.

### *5.2. Static Test*

At first, the result of the collaborative-learning is compared to the simple constant damping control. According to Figure 7, the two best constant damping controls and the CL-network are performing similarly well. The differences are 0.3% and therefore irrelevant. We explain this as the neural network is not strong enough to clearly identify the different sea states; this corresponds to the small variability of the damping coefficients. Instead the CL-network is able to find a good working point for all sea states, that it varies slightly.

### *5.3. Dynamic Test*

While the static test shows that the network is able to get the most beneficial damping from existing controls, the dynamic run shows that a group of CL-algorithms is able to find (local) optimums without any a priori information. Moreover, the results in Figure 7 show that the CL-ANN1 and CL-ANN2 push each other to achieve better results and so absorb even more power than during the dynamic test. The second CL-network is at a disadvantage because of the large exploration factor. In real applications this factor would be much slower; however, in this case a large exploration factor helps to accelerate the learning process. The advantage of the CL-ANN1 control over the constant damping WECs is with 1.6% very small. This must be seen in relation to the influence of the damping factor on the absorbed power: the ratio between the highest and lowest constant damping factor is 2.5, but the maximal difference in absorbed power between two constant damping WECs is only 7.9%. Huge differences in power absorption in the evaluation wave sequence are therefore not expected with this WEC type.

### *5.4. Wave Tank Test*

During the test under realistic conditions in the wave tank the network showed no problems with handling noisy real world data as can be seen in Figure 8. The damping coefficient is fluctuating a

bit, but all in all the ANN follows the reference values very well. This is not necessarily a bad sign, it shows that the system is not over trained and therefore has not lost its ability to generalize.

## 6. Conclusions & Outlook

In this paper the CL learning was introduced which is a very flexible way of controlling a converter compared to traditional supervised or ensemble learning strategies: The process does not have to be divided into training and operation state, as it does in traditional supervised or ensemble learning; Instead the learning is done "on the fly" during normal operation.

When using several CL-networks with different characteristics, this can lead to a dynamic process, in which the networks will "push" each other to improve the parameters. This control strategy is especially useful in high dynamic environments with interactions between the controlled units.

The control did not improve the absorbed power significantly. The artificial neural network used in this model was not strong enough to handle the noisy data and classify the sea states clearly. Ways to tackle this problem could be a very deep network or an algorithm easing the learning process by preprocessing the data; for example by splitting the learning process into wave based sample sets or by analysing the sea state and using wave height and energy period as inputs of the network.

Controlling only the damping of an energy converter has a minor influence compared to many other control strategies. This can be noticed in the small differences between the different damping coefficients in Figure 7. Especially latching and reactive control can increase the power absorption significantly. To implement the CL network for one of these controls could ease the search for the optimal parameters for the network, compared to the damping control with its very small differences.

This paper neglected the interaction between the WECs in line with the wave crest, assuming these will have minor influence on the optimal damping as for example the results of [21] suggest. Further studies could focus on the influence of interaction between WECs - first on the interaction of WECs in a line parallel to the wave, then also interaction between the WECs in several rows.

**Author Contributions:** The general conceptualization and methodology were done by S.T. who also wrote most of the paper. The conceptualization and design of the physical PTO were done by J.E., M.E. and S.T. The planning of the wave tank experiments were done by M.G. (Marianna Giassi) and S.T. with help of J.E., M.G. (Malin Göteman), E.R. and M.H. The simulation tool was written by S.T. M.G. (Marianna Giassi), M.G. (Malin Göteman), J.E. and J.I. were involved in all stages of the project and contributed with ideas and advice. Project administration, including supervision, was done by M.G., J.E., M.E., M.H. and J.I.; M.G., J.E. and J.I. were furthermore responsible for the funding acquisition. All authors contributed to the paper with reviewing and editing.

**Funding:** The authors want to thank the Swedish Energy Agency (project number 40421-1), the Swedish Research Council (VR, grant number 2015-04657) for funding this research and the Åforsk Foundation. This work was supported by Stand Up for Energy.

**Acknowledgments:** This paper would not exist in this form without the help of Oliver Goldsmith, Liz Dunsmoor, Tara Büttner and Keiran Monk during the wave tank test. Furthermore the authors want to thank the Swedish Energy Agency (project number 40421-1), the Swedish Research Council (VR, grant number 2015-04657) and the Åforsk Foundation for funding this research. This work was supported by Stand Up for Energy.

**Conflicts of Interest:** The authors declare no conflict of interest.

## Abbreviations

The following abbreviations are used in this manuscript:

| | |
|---|---|
| ANN | Artificial neural network |
| CL | Collaborative learning |
| CL-ANN1 | 'deep' ANN used for the learnable WEC |
| CL-ANN2 | 'shallow' ANN used for the learnable WEC |
| COAST laboratory | Coastal, ocean and sediment transport laboratory; |
| | Facility at the University of Plymouth containing the wave tank |
| PTO | Power take-off |
| WEC | Wave energy converter |

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
