# Peer review of "A Model Free Control Based on Machine Learning for Energy Converters in an Array"

_2504-2289, doi:10.3390/bdcc2040036_

Reviewer 1 Report

title: the title does not convey the meaning of what is done. we see a model (7) but the title says it is a model free ... Therefore, please consider much better and honest expression.

abstract: please write abstract in one paragraph. we don't understand the scope and limitation of the work from the writing. where is the originality? please explain.

introduction: the authors here wanted to say that we can develop a model of energy conversion system (e.g., wave energy conversion system) using an analytical approach. alternatively, we can machine learn some hidden structure in the data (e.g., in the data of surface elevation of a wave energy conversion system) which serves the purpose of the model. machine learning hidden structures require a computational intelligence based system (e.g., a system based on ann).

however, employing ann or other machine learning method is also a cumbersome process because those who are working in this area for long time, we know that it is difficult to set the ann for given problem.

please revisit your arguments.

figure 1 does not convey any message. it can be removed.

section 2. section 2 is bit confusing. the descriptions are very brief and convey no meaning to general readers.

it is better to present the model of the wave energy conversion system (model in figure 7) in section 2 to and make the premise of the paper. then in section 3 (now section 2) the authors can describe the computational framework.

the point the reviewer wants to make is that still the logical order of the contents are not organized the way that it should be. the authors must redo the manuscript to make it publishable.

Author Response

Dear reviewer,

thank you for the time you spent reading and reviewing our article and providing feedback. In the following we want to respond to your comments in detail:

First of all we want to point out the focus of this paper that is not providing something substantial new to the field of machine learning, but focus on how to implement machine learning elegantly in a specific real world application. The core idea of the paper is that while using the assumption made in Figure 1, several control strategies can be evaluated at the same time in the same situation and so we can easily produce a labeled data pair matching input data (state) with the best action (out of the chosen actions). The learning concept itself is nothing groundbreaking new, more a "low-level" machine learning method. But the transfer of this simple approach in the presented real world application is new. State of the art work, as cited in the paper, focuses only on reinforcement learning of single energy converters with complex reward functions. But when looking on arrays and assuming this very special case where several identical energy converters are effected by the same input at the same time, this simple approach can be used.

We are aware that paper like this one are rare in journals like BDCC, but we want to contribute to the interdisciplinary exchange and while we were reading a lot of machine learning paper, also contribute to the discussion how machine learning might be implemented in different fields and what are the specific challenges in each field.

"the title does not convey the meaning of what is done. we see a model (7) but the title says it is a model free ... Therefore, please consider much better and honest expression."

I am afraid, here is a misunderstanding: model-free control means, that the control algorithm is not relying on a pre-defined model, so it will adapt to whatever it has to control. This is a very important aspect, as for many energy converters - due to friction and other "noise" - the exact parametrisation is not known, and was the reason for us to use a machine learning approach. The evaluation is done with a model, but the controller itself has no knowledge about the model.

Even so the title is still the same, we changed the abstract slightly, hoping that it better understandable.

"however, employing ann or other machine learning method is also a cumbersome process because those who are working in this area for long time, we know that it is difficult to set the ann for given problem."

I agree, but also want to add that one don't have to work in this area for a long time to know how difficult it is to set the ANN for a given problem. And as can be seen in our results, the ANN is far away from being perfect. However, as mentioned above the method is the main focus of this work.

"figure 1 does not convey any message. it can be removed."

Figure 1 was aimed to show how a wave energy converter array suitable for the advertised control has to look like. It might be banal, but on the other side, at least for wave energy converter, the optimal array design, the use of identical and differently shaped energy converters and the interaction between the energy devices is an on-going field of interest (more information can be found in: Giassi, M., Multi-parameter optimization of hybrid arrays of point absorber Wave Energy Converters). This figure summarizes all assumption that allows us to transfer the complex problem of controlling an energy converter array into the simple the simple machine learning approach for this problem. We think therefore that it is still very important, please let us know if you still thing it should be removed, or if you have ideas how to improve the picture.

"section 2. section 2 is bit confusing. the descriptions are very brief and convey no meaning to general readers."

The (self-written) ANN algorithm had to fit in the existing experimental set-up (in planned further tests we want to use the ANN during the wave tank test run) and as the complete set-up was very complex we decided to keep the machine learning as simple as possible. We now added a paragraph right before the subsection "Evaluation" explaining the few addition we made on the standard multi-layer-perceptron network.

"it is better to present the model of the wave energy conversion system (model in figure 7) in section 2 to and make the premise of the paper. then in section 3 (now section 2) the authors can describe the computational framework."

We had the discussion in the author team, and decided for this order. The principal aim of this work is to introduce the general idea, because similar array layout can be found in different sustainable energy plants (wind power farms, (further) hydro current farms, ...). So we wanted not only focusing on wave power.

Inspired by your comment, we decided to make this border clear, splitting the "Methods" (section 2) section into "Collaborative Learning" and "Numerical and experimental set-up" (new section 3). So up to section 3 we present the general approach before we focus on wave energy converter.

We hope we could clarify some points. If you still disagree in some points please let us know. This is our first paper in a machine learning journal and as every disciplines emphasis different aspects, we are curious to hear how we can improve our article for the audience of this journal.

Reviewer 2 Report

I think that the work done is enough for publication. Therefore, the paper can be published in present form. I can not provide helpful comments since the manuscript is very interesting and well written.

Author Response

Thank you very much for reviewing our paper. We are glad to hear that you enjoyed reading it.

Reviewer 3 Report

Figure 8: contant damping absorbes 99.7% of energy then why is CL-ANN1 performing particularly better?

Please provide more details on the implementation, i.e. software used, if it is publicly available to be used by people, data and code both?

Author Response

Dear reviewer,

thank you very much for spending your time and reviewing our paper. We appreciate your effort and want to comment on your feedback in detail:

"Figure 8: contant damping absorbs 99.7% of energy then why is CL-ANN1 performing particularly better?"

The 0.3% differences between best constant damping and CL-ANN1 is only for the run where the CL-ANN1 is trained with the constant dampings. When trained together with CL-ANN2, the CL-ANN1 network absorbs 101.3%. even so an increase in power absorption of 1.6% is a small differences, it shows at least that the CL-ANN1 is able to improve the power absorption.

"Please provide more details on the implementation, i.e. software used, if it is publicly available to be used by people, data and code both?"

The software is self-written in order to integrate seamless and with good performance in the existing WEC control framework. It is therefore very elementary. We added a small description before the subsection "Evaluation". The strong link with the existing WEC control ecosystem was also the reason to not make the code public available. However, due to the fast development of ANN frameworks, we are thinking of using an established ML-library for future work.

Round  2

Reviewer 1 Report

authors have improved the manuscript but still needs revisit.